# Evaluating the Feasibility and Acceptability of a Prototype Hospital Digital Antibiotic Review Tracking Toolkit: A Qualitative Study Using the RE-AIM Framework

**DOI:** 10.3390/antibiotics14070660

**Published:** 2025-07-01

**Authors:** Gosha Colquhoun, Nicola Ring, Jamie Smith, Diane Willis, Brian Williams, Kalliopi Kydonaki

**Affiliations:** 1School of Health and Social Care, Edinburgh Napier University, 9 Sighthill Court, Edinburgh EH11 4BN, UK; n.ring@napier.ac.uk (N.R.); j.smith7@napier.ac.uk (J.S.); d.willis2@napier.ac.uk (D.W.); 2Institute of Health Research and Innovation, University of the Highlands and Islands, Old Perth Road, Inverness IV2 3JH, UK; brian.williams@uhi.ac.uk; 3Department of Medicine, University of Athens, 75 Assias Street, 115 27 Athens, Greece; ckydonaki@nurs.uoa.gr

**Keywords:** antimicrobial resistance, antibiotic review, behaviour change, optimisation, person-centred research, user experience, qualitative design

## Abstract

**Background:** Internationally, digital health interventions have increasingly been adopted within hospital settings. Optimising their clinical implementation requires user involvement, but there is a lack of evidence regarding how this should be done. **Objectives:** This study was carried out to understand the acceptability and usability of a prototype Digital Antibiotic Review Tracking Toolkit and identify modifications required to optimise it ahead of a trial. **Methods**: The optimisation process involved online semi-structured interviews with a purposive sample of fifteen healthcare professionals recruited from Scotland and England, along with three service users, to gather feedback on the prototype’s design, content and delivery. Participants’ negative views were specifically sought to identify adaptations needed to ensure that the intervention’s components aligned optimally with end-user needs. Data were analysed using Framework Analysis guided by the RE-AIM implementation science framework (Reach, Effectiveness, Adoption, Implementation, and Maintenance) to identify key themes. **Results**: Participants mostly voiced positive views regarding the prototype, finding it acceptable, feasible and engaging. They also identified concerns relating to its adoption, system functionality, accessibility and maintenance that needed to be addressed. Anticipated low adoption rates were linked to issues surrounding computer literacy. This detailed user feedback informed rapid adjustments to the intervention to enhance its acceptability, perceived future credibility and usability in hospitals. **Conclusions**: This novel study illustrates how to identify, modify and adapt a digital intervention quickly and efficiently using qualitative iterative methods. Findings highlight the critical importance of contextualising end-user experience with health interventions to facilitate future engagement, uptake, and long-term use. This study also demonstrates how core elements of the MRC framework can be operationalised to help refine prototype digital interventions pre-trial.

## 1. Introduction

Antimicrobial resistance (AMR) is a global health crisis, projected to cause 10 million deaths annually by 2050 if left unaddressed [1]. Inappropriate and prolonged antibiotic use in hospitals is a major contributor to this problem, where clinician uptake and timely antibiotic review are key strategies in antimicrobial stewardship. Yet, the uptake of review processes remains inconsistent and suboptimal across hospital settings [2,3].

The careful development of complex interventions, often involving clinician behaviour and/or service organisation change, is required to generate improvements in the quality of healthcare and patient outcomes [4,5]. However, the design, evaluation and implementation of such interventions often present practical and methodological challenges due to their multiple interacting components, the intensity of behaviour change required by those delivering or receiving the intervention, the number of groups, settings or levels targeted, and the interaction with continually changing clinical contexts [4,6].

Methods to improve intervention development and reduce implementation failure have significantly advanced over the past two decades, supported by frameworks such as the Medical Research Council’s (MRC) guidelines [4,7]. These frameworks stress the importance of addressing uncertainties related to intervention acceptability and delivery before moving to large-scale trials. This includes refining intervention content and assessing implementation strategies to ensure these are suitable for the characteristics of the target population and can be delivered within the existing skills, resources and infrastructure of provider organisations. This refinement requires pre-trial optimisation, which is an iterative and data-driven process [8]. Such optimisation can be framed as a formative evaluation, using structured implementation science frameworks (e.g., RE-AIM —Reach, Effectiveness, Adoption, Implementation, and Maintenance) to guide the improvement process [9]. This purposeful intervention adaptation routinely occurs in other fields, such as engineering and information technology (IT), where early engagement with the end-users facilitates incremental improvements and leads to the optimisation of the product’s performance but has been under-used to date in health research [10,11].

While interest in adapting existing interventions has grown, less attention has been given to optimising new health interventions prior to trials. Yet, routinely refining health interventions could reduce research waste and create a more responsive, feasible evidence base [12]. This gap is particularly relevant in the context of rapidly evolving antibiotic-related digital health technologies (DHT), where randomised trials risk becoming obsolete by the time results are published [13,14]. Furthermore, the effectiveness of these interventions is often unclear due to the limited description of the ‘mechanisms’ behind their success or failure [10]. Specifically, there remains a paucity of evidence on their design, usability, stakeholder involvement during development, and integration into routine clinical workflows [1,15].

The Digital Antibiotic Review Tracking Toolkit (DARTT) is a prototype digital intervention designed to support the initiation and documentation of antibiotic ‘time-outs’—that is, timely reviews of antibiotic use—in acute hospital settings. DARTT comprises four components: an Antibiotic Tracker, webinar, e-learning module, and patient information materials (see Section 4 and Appendix A for a full summary of DARTT components). During initial prototype development, it is essential that early end-user feedback is sought if developers are to ensure digital interventions are acceptable and engaging and that such feedback is responded to if they are to overcome issues likely to affect future uptake and adherence [16]. Although IT developers routinely identify and specify the needs of individuals expected to use new applications within an organisational context, there is limited detailed guidance on how to effectively accomplish this in healthcare research [11]. One recommended approach in healthcare is to conduct small qualitative studies designed to target discrete research questions that aim to uncover how the intervention can be redesigned to better suit the target population [16,17]. Qualitative research, as a means of obtaining and incorporating user perspectives, can, therefore, enable rapid optimisation of digital interventions while ensuring intervention development and refinement retain all the elements that theory and evidence suggest will be effective in supporting behaviour change [18].

In the specific context of hospital electronic Medication Management Systems (eMMS), recent studies have highlighted several barriers to their adoption. For example, clinician ‘passive resistance’—characterised by superficial compliance without active engagement with the eMMS—has been identified as a significant challenge to successful implementation [19]. This study also showed that user hesitance can impede technology adoption and emphasised the need for a better understanding of sociotechnical dynamics when considering intervention implementation. Zhou et al. [20] employed configuration theory to assess health information systems, revealing that the success of eMMS depends on aligning human, organisational and technological factors with the unique characteristics of healthcare institutions. Wu et al. [21] further examined how factors influencing eMMS benefits evolve over time, indicating a need to better understand the long-term impacts of these systems as institutions adapt. Lastly, a more recent study provided evidence of ongoing challenges and partial successes in eMMS implementation, highlighting the need for longitudinal research on technological adaptation and user acceptance in practice [22]. Overall, these studies highlight the potential of qualitative research to enhance the evidence base by enabling intervention developers to gain a comprehensive understanding of user perceptions and improve the usability and adoption of health IT prior to trials, whose adoption remains low in acute hospital environments [23].

This paper reports on the formative evaluation of the DARTT prototype using the RE-AIM framework to guide its optimisation [9]. The work is part of a larger programme of intervention development [24,25]. It introduces an original approach to digital health intervention development, applying person-centred design in antimicrobial stewardship within hospital settings—an area of critical international importance. This formative evaluation focused on how DARTT could be refined to enhance its feasibility, acceptability, and usability before being tested in a future trial.

### Study Purpose

The study objectives were to (1) identify user-reported factors affecting the feasibility, acceptability, and usability of the DARTT prototype intervention across the RE-AIM domains and (2) determine any user-informed modifications needed for the design, content, and delivery of DARTT. Together, these objectives aimed to guide the optimisation of DARTT to align with stakeholder preferences, support integration into real-world clinical workflows, and address known challenges in conducting timely antibiotic reviews in hospital practice.

## 2. Results

The results are presented first as themes reflecting Objective 1, followed by an overview of how these insights informed subsequent modifications to the DARTT intervention (Objective 2).

### 2.1. Key User-Reported Themes on the DARTT Intervention

Four themes generated from the data are described below. Participants were generally positive about the intervention and could see its clinical value: ‘*I think it’s excellent, very logical and [seems] easy to follow*’, taking account ‘*of the various people at various stages involved in monitoring and administering antibiotics*’ [P18, Health Service User]. However, this paper focuses predominantly on participants’ negative reactions to using the intervention to illustrate the adaptations needed to enhance its applicability in real-world settings.

#### 2.1.1. Theme 1: Tailoring System Functionalities and Design

While all participants endorsed DARTT, for example, stating that ‘*it’s striking how useful the functionality is, there’s nothing like this at the moment*’ [P3, Consultant Microbiologist], they also identified many aspects of the intervention that would benefit from refinements, either directly or by inference. Prominent among participants’ comments was how the intervention design and features could be tailored in different ways to suit the user. They suggested many practical adaptations, including the need to ensure that the Antibiotic Tracker (Component 1) interface is intuitive and user-friendly and that the design fits well into the clinical workflow. Several senior healthcare professionals (HCPs) voiced concerns that incorporating multiple data fields into the Tracker could potentially be excessively time-consuming in a busy hospital setting, for example:


*“There are too many data fields, and I’m not sure if having all of those questions…is worthwhile.”*
[P5, Infectious Diseases Consultant]

Other practical improvements focused on clinical decision support. Participants commonly reported cognitive overload and raised concerns about the volume of irrelevant alerts, suggesting these should be reduced and prioritised based on clinical relevance. They discussed the need to ensure instant access to antibiotic prescribing guidelines and offer guidance on recommended therapeutic options for the condition being treated. Creating user authorisation codes for restricted antibiotics was seen as helpful.


*“So, in another [Health Board], antibiotics are ‘locked down’ to a much greater degree, and you need an authorisation code from the on-call microbiologist to prescribe.”*
[P11, Clinical Pharmacist]

When viewing the main menu page, participants also identified some ‘aspirational’ adaptations, including many desired software features and interface requirements, such as individualised dashboards, pop-up windows for reviewing microbiology results and outstanding action reminders. Among other modifications identified was an option for users to review prior activities and embed an online calculator.


*“Could you have a calculator on another tab for Vancomycin and Gentamicin? Then you can click it and calculate [dosage] on the Tracker.”*
[P12, Advanced Nurse Practitioner]

#### 2.1.2. Theme 2: Bridging the Technology Gap

Some participants identified technological infrastructure and system compatibility as potential issues in implementing DARTT. They expressed concerns about system configuration (e.g., wireless infrastructure) as well as interfaces with existing systems (e.g., paper-based prescription charts). Participants commented that the Antibiotic Tracker would need to integrate into existing routines and practices: ‘*Having information moved from one system to another requires lots of harmonisation. It could only work if it’s integrated and coded into current practice’* [P5, Infectious Diseases Consultant]. Some described their previous experiences of using obsolete and outdated technology.


*“IT [Information Technology] in the NHS generally doesn’t work for what you need. The computer in my office is 20 years old! The problem is also Wi-Fi. We have upload speeds of like 1.”*
[P15, Infection Surveillance Nurse]

Anticipated low adoption rates were linked to issues surrounding computer literacy and a general apprehension about electronic prescribing. There was a prevailing perception that certain HCPs may not feel entirely at ease with IT systems.


*“There’s a massive fear around electronic prescribing. I work with a lot of practitioners who have IT issues and find it difficult to navigate online.”*
[P14, Advanced Nurse Practitioner]

Participants further noted that the intervention uptake required evaluating the training needs of all types of users who will interact with the Antibiotic Tracker. However, participants reported mixed views related to challenges in delivering practical training to large numbers of staff. Many participants liked the idea of completing training at home, away from the busy hospital. Healthcare professionals preferred interactive or hands-on learning to familiarise themselves with and practice using the system and suggested that a video, including a ‘run through’ demonstration of DARTT, would be helpful.


*“I recently did my [NHS prescribing] online training and found the video training really useful, especially for seeing the functionality.”*
[P2, Consultant Physician]

Some participants commented that technical expertise was needed to customise systems and provide sufficient support for DARTT users. They further emphasised the need to provide a selection of tools to facilitate implementation and ongoing IT assistance to help with any technical difficulties and ensure the new functionalities are adequately used.


*“You need a named person to contact for advice, so if I come across something on the Tracker I’m not sure about, I can call for advice and say: ‘How do I deal with this?’.”*
[P17, Health Service User]

#### 2.1.3. Theme 3: Maintaining Organisational Leadership

Participants reported that setting up a project management team to coordinate future work processes was essential to support implementation. They felt that ongoing high-level commitment from key stakeholders was required to ensure that sufficient resources were devoted to supporting the adoption and implementation of DARTT. Participants discussed issues related to active engagement, including the need to gather support from the decision-makers.


*“Don’t just put it out and expect it to be taken up—it won’t be. It needs to be engaged with and sold.”*
[P15, Infection Surveillance Nurse]

They commented that such organisational leadership needs to drive system improvements continuously and consistently by following the transformational vision and determining medium- and long-term priority areas. Linking DARTT to Antimicrobial Stewardship Teams within the hospitals and engaging enthusiastic senior clinicians to provide leadership also featured highly as an example of organisational leadership.


*“If the medical director, director of nursing and director of pharmacy all say, “We support this”, people are more likely to use it.”*
[P3, Consultant Microbiologist]

Engagement from frontline staff was considered just as important as leadership support. As one consultant physician noted: ‘*It has to be both, engagement from the top–down and from people like junior doctors and nurses who are physically there. You can’t succeed without it*’ [P2]. Likewise, all lay participants highlighted the importance of involving patients in treatment decisions, including antibiotic use, with trust, communication, and relationship-building seen as central to person-centred prescribing. As one health service user put it:


*“Engage patients as well—that helps build trust and endorse the value of the whole project.”*
[P18, Health Service User]

Approximately half of the HCPs believed that implementing DARTT would be relatively uncomplicated and would likely face minimal resistance. One clinical pharmacist described the situation as *‘pushing against open doors*’ [P9], suggesting that introducing the intervention would be straightforward due to strong support from Antimicrobial Stewardship Teams and clinicians’ readiness to adopt it, along with the absence of significant barriers.

Others were more sceptical, expressing concerns that deeply ingrained habits and resistance to change could potentially hinder the implementation process. To overcome these barriers, participants suggested a quick but direct summary with the rationale for DARTT embedded within routine team meetings. They also suggested shortening the webinar (Component 2) to make it easier to view alongside clinical demands, with most preferring it to be available both live and as a pre-recorded session to encourage greater participation.


*“Maybe having two versions, a shorter and a longer one, would work better. You could pre-record it and play it during staff meetings.”*
[P10, Clinical Pharmacist]

#### 2.1.4. Theme 4: Lessons Learned and Sharing of Experiences

Participants felt that progress monitoring was central to the maintenance of DARTT, including identifying issues and *‘easing out the glitches’* [P1, Consultant Physician]. There was recognition that regular evaluation would be essential to assessing anticipated benefits and facilitating learning. Participants agreed that future system iterations based on user feedback, including software upgrades and functionality improvements, were crucial for further buy-in and dissemination of innovative ideas.


*“Once people start to use it, get their feedback on benefits and share with others… and if something isn’t right, review and change it.”*
[P11, Clinical Pharmacist]

However, participant views relating to receiving personal DARTT prescribing feedback were mixed. Although some welcomed the idea of creating feedback mechanisms, others pointed out that regular face-to-face feedback on prescribing could be perceived negatively. There was a perception that if the feedback offered is unsolicited, ‘*some prescribers might feel persecuted*’ [P8, Medical Trainee]. Some clinicians raised concerns about providing prescribing feedback to users based on comparative data, citing that variations in practice across different specialities could influence local prescribing patterns.


*“Individual feedback has to be contextualised and fairly diplomatic, because if it’s not done carefully, it could be perceived as criticism.”*
[P7, Resident Physician]

Instead, they suggested providing unit- or hospital-level feedback that could be discussed in team meetings. Keeping in contact with end-users, as well as providing hospital managers with regular up-to-date prescribing information during staff meetings, emerged as key.


*“It would be good to present some data at team meetings, and compare performance to other hospitals, just to keep it fresh and maintain the benefits.”*
[P9, Clinical Pharmacist]

In the context of this study, participants were actively encouraged to share negative experiences. Clinicians emphasised the importance of performance follow-up and the ongoing identification and resolution of unintended consequences or DARTT system errors. Several participants suggested strategies for this, including data quality monitoring and clinical audits.

### 2.2. User-Reported Themes Related to the DARTT Prototype Mapped to RE-AIM Domains

The thematic analysis identified four key themes: (1) Tailoring System Functionalities and Design, (2) Bridging the Technology Gap, (3) Maintaining Organisational Leadership, and (4) Lessons Learned and Sharing of Experiences. These themes correspond to the RE-AIM framework domains [9]. Theme 1 reflects *Effectiveness*, highlighting user-driven refinements that improve user-friendly system design and usability. Theme 2 relates to *Reach* and *Implementation*, emphasising the importance of infrastructure compatibility and training to enable equitable access and integration into existing workflows. Theme 3 aligns with *Adoption*, highlighting the critical role of leadership engagement and organisational support for successful uptake. Theme 4 connects with *Maintenance*, capturing insights about ongoing evaluation, long-term use, and iterative improvements. Table 1 illustrates how interview themes related to DARTT acceptability and feasibility map onto the RE-AIM framework.

### 2.3. Table of Changes: User-Informed Modifications to DARTT

Participant interview responses primarily focused on the content, functionality, and design of the DARTT prototype. These user-informed insights guided rapid, ongoing adjustments to optimise the intervention. Suggested changes were reviewed by the research team and then retrospectively compared with the key behavioural deficits identified in Phase I of the study (intervention planning and development) using the Behaviour Change Wheel [25,26] before being implemented as adaptations to the DARTT prototype (see Appendix A for further details).

The main user-informed modifications made to DARTT during this study involved the Antibiotic Tracker’s functionality (Component 1). For example, researchers improved the layout and navigation of the DARTT dashboard, reduced unnecessary user alerts, and added direct links to prescribing guidelines. Table 2 presents the key intervention changes based on participant feedback, focusing only on DARTT Components 1–3, as the Patient Information Materials (Component 4) were reported as highly acceptable by participants and required no modifications.

## 3. Discussion

This paper is part of a series reporting on the development and refinement of the DARTT intervention (see Figure 1), complementing earlier work on establishing a robust theoretical basis for the intervention [24,25]. This study illustrates how qualitative research methods were used to capture user-informed feedback and rapidly optimise the prototype DARTT intervention through stakeholder interviews with HCPs from Scotland and England involved in antibiotic prescribing and health service users. Although DARTT has yet to be tested, the incorporation of person-centred design principles in soliciting feedback on DARTT content has enhanced its potential for greater acceptance among its target audience, which is needed to support future implementation [17].

Integration with existing hospital workflows and electronic health record systems (EHRs) is essential for DARTT’s successful implementation. Integration could be achieved by ensuring compatibility through standardised Application Programming Interfaces (APIs), facilitating seamless data exchange between DARTT and hospital EHR systems. Training resources would need to be developed to familiarise staff with new processes, alongside ensuring ongoing IT support. Additional resources for integration would likely involve technical expertise, infrastructure upgrades, and software customisation, with financial implications such as initial development costs, staff training, and ongoing maintenance needing careful consideration and planning [20,21].

The implementation of health IT solutions, such as electronic health records and electronic medication management systems, holds transformative potential for patient care and system efficiency [27]. Yet, these efforts often face obstacles due to complex technical, human, and organisational factors. A key barrier is user resistance—both passive (e.g., low uptake) and active (e.g., refusal to engage) [28]. For example, Kim et al. [19] linked passive resistance to concerns about autonomy, poor incentives, and lack of training. These themes echoed our findings, where safety-focused prescriber feedback was sometimes perceived as criticism. Addressing these challenges through early engagement, tailored training, and aligning system goals with user priorities can improve adoption [29]—a crucial step for successful digital interventions to support responsible antibiotic use globally [1].

Successful health IT implementation requires balancing human, organisational, and technological factors [1,20]. Collaboration amongst interdisciplinary HCPs, supportive policies, and user-friendly systems are all key [20]. Customising system configurations to meet the organisation’s needs, rather than applying a one-size-fits-all approach, also enhances implementation processes [1,30]. Leadership backing and adequate resources further support acceptance and integration [31]. This qualitative study provides insights into achieving this balance to enhance antibiotic review safety and quality in acute hospitals, such as by focusing feedback at the unit or hospital level. Protecting patient data within DARTT is a critical consideration requiring strict UK data protection compliance, encrypted transmission, secure NHS-approved storage, and audit trails ensuring accountability and transparency.

Health IT systems like eMMS must evolve as users adapt and workflows change, with initial disruptions often giving way to greater efficiency and safety [21]. Continuous, user-driven improvements are essential to keep systems effective, though this demands significant time and resources [27,32]. Kim et al. [22] highlight that user satisfaction and system effectiveness evolve with updates and organisational support, making regular evaluation and adaptation vital. Striking the right balance between fast deployment and thorough design with meaningful user engagement is key to long-term success [31].

Whilst participants praised many aspects of the DARTT prototype, they also raised concerns about system integration, training needs, and resistance to change. Simplifying the Antibiotic Tracker (Component 1) to better align with clinicians’ workflows was identified as a key adaptation priority. Both clinicians and health service users emphasised the need for an intuitive, seamless system with minimal technical disruptions—echoing previous studies where HCPs favoured simple, time-efficient tools for antibiotic review [32]. A key difference in perceptions was the unique emphasis by health service users on person-centred prescribing, highlighting the importance of incorporating patient involvement in future antibiotic interventions.

Concerns about computer literacy and electronic prescribing emerged during interviews, with participants stressing the need for effective training to use the Antibiotic Tracker confidently. Mandatory, interactive online training was seen as crucial to building competency and increasing DARTT uptake. Although evidence on optimal e-prescribing training is limited, web-based methods have proven effective for educating large healthcare groups [33]. DARTT feedback mechanisms directed at prescribers elicited mixed reactions; some participants expressed concerns that regular individualised feedback might feel socially punitive, aligning with earlier literature describing a culture in which junior clinicians tend to avoid confrontation with senior colleagues [24]. To address this, the intervention incorporated positive feedback strategies, such as sending emails to prescribers highlighting instances of good practice, aiming to encourage learning and motivation without inducing fear [34]. Findings also emphasise the need for securing both top–down buy-in from senior decision-makers and bottom–up engagement with frontline staff and patients for successful digital intervention adoption. This supports the view that lack of sufficient engagement often leads to project failure [35]. Ensuring DARTT’s uptake will require involving key multidisciplinary stakeholders through awareness-raising (e.g., staff meetings), fostering local ownership, and securing support from management and end-users.

Finally, the literature increasingly acknowledges the importance of ongoing progress monitoring, evaluation, and systems adaptation to realise anticipated benefits [31,36]. Adaptation is central to DARTT’s design, supporting its translation into practice and long-term use. While there was general optimism amongst participants about DARTT’s potential for self-sustainability, ongoing evaluation was identified as a crucial design feature to support its implementation across diverse hospital contexts. Evidence indicates that fully functioning technology can improve productivity, enhance patient care, and transform NHS staff workflows [35]. However, if end-user feedback is not actively addressed, the system’s value is unlikely to be fully realised [18,37]. Key strategies for ensuring DARTT’s sustainability include long-term follow-up (e.g., through data quality monitoring and clinical audits) and system iterations (e.g., software upgrades and functionality improvements) informed by end-user feedback. Identifying unintended consequences and system errors that may not have been apparent during initial testing aligns with MRC guidelines for intervention development [4,7].

In conclusion, sustainability and adaptability are essential for health IT systems to remain effective in dynamic healthcare settings [38]. Flexibility is necessary to respond to evolving user needs, regulatory changes, and advancements in healthcare delivery. Ongoing updates, training, and performance evaluations are key to ensuring continued system relevance and benefits [31]. Successful health IT implementation relies on strategic planning, user engagement, and a commitment to continuous adaptation. By addressing resistance, aligning solutions with organisational needs, and fostering iterative improvements, healthcare organisations can maximise the impact of health IT systems on patient care and clinical outcomes.

### 3.1. Strengths and Limitations

This study was conducted during the COVID-19 pandemic, when face-to-face data collection was suspended, requiring a rapid shift to virtual methods. This unexpectedly enabled more diverse recruitment of HCPs and lay participants across the UK, facilitating the capture of a wider range of views, which in turn supported data saturation and enhanced the transferability of findings. Using a simple DARTT prototype (see Section 4) allowed rapid insights into users’ needs and priorities, fostered meaningful dialogue, and helped refine key intervention features. Although participants were knowledgeable and engaged in antibiotic prescribing, less experienced HCPs were under-represented. As such, the sample likely reflects individuals with a strong antimicrobial stewardship interest and a greater likelihood of engaging with prescribing interventions. Another limitation was the interviewer’s dual role as both DARTT developer and data collector, which may have influenced participants to provide socially desirable responses. However, interviewees still offered critical feedback on the prototype.

The study explored HCPs’ perceptions of the prototype within acute care settings across the Scottish and English NHS; however, it did not assess real-world use, limiting understanding of its practical feasibility and effectiveness—areas that warrant further research both within the UK and internationally. Refining the intervention through qualitative feedback involved several iterations, making scaling resource-intensive (e.g., time, finances, and human resources) and complex. The economic evaluation was not conducted. Rapid qualitative methods focused on immediate feedback but might have missed deeper, long-term user insights. The RE-AIM framework guided systematic data collection and analysis, while themes emerged organically and were refined through multidisciplinary consensus. Collaboration with NHS Trusts and input from microbiology, infectious disease, and behaviour change experts helped ensure accurate interpretation of data and emerging themes, mitigating researcher interpretation bias.

### 3.2. Future Implications

Study findings have implications for various groups. Policymakers can utilise the detailed software features identified by users to enhance existing e-prescribing systems and inform national strategies for antimicrobial stewardship. Clinicians can leverage these insights to critically evaluate the usability, acceptability, and integration potential of digital tools within hospital workflows. Researchers can use findings and this approach to guide precise intervention development and optimisation strategies in various clinical contexts.

The next steps include conducting a structured feasibility trial to evaluate DARTT’s practical use in acute clinical settings across the UK, focusing on intervention fidelity, usability, and preliminary indicators of effectiveness, such as changes in antibiotic prescribing. Key outcome measures should cover user satisfaction, intervention usage, guideline adherence, and documentation accuracy. Broader stakeholder engagement will involve multidisciplinary advisory groups and patient panels to refine DARTT during testing. Future phases should incorporate cost-effectiveness analyses and clear scale-up strategies, with a phased roll-out across hospitals supported by continuous stakeholder feedback to enable iterative adaptation and wider acceptance.

## 4. Materials and Methods

To enhance transparency and reporting rigour, the study followed the Consolidated Criteria for Reporting Qualitative Research (COREQ) guidelines [39]. Trustworthiness was assessed using Lincoln and Guba’s criteria [40]: credibility was ensured through pre-tested topic guides, triangulation, peer debriefing, and team validation; transferability through purposive sampling; dependability via inductive analysis, detailed documentation, and reproducibility checks; and confirmability through reflexive journaling, independent coding, and team review. Full methodological details are provided in the Appendix A.

### 4.1. Design

DARTT development was two-phased; this paper reports on Phase 2. In Phase 1 (see Figure 1), researchers used findings from their systematic review and meta-ethnography [24] with exploratory focus groups of key stakeholders. These data were then triangulated using the Behaviour Change Wheel to identify key mechanisms of behaviour change and relevant intervention components and reported separately [25,26]. The intervention design was informed by behaviour change theory, specifically the COM-B model and Behaviour Change Technique Taxonomy, which provide a comprehensive, evidence-based method for linking interventions to mechanisms of action, supporting systematic development of behaviour change strategies [26,41]. The aim was to ensure the DARTT intervention is acceptable and relevant for those who will use it while retaining theory and evidence-based elements supporting behaviour change. A process of behavioural analysis then identified likely core effective intervention components, which needed to be incorporated into the prototype intervention.

Phase 2, reported in this paper, involved a rapid inductive optimisation process. The researchers conducted one-to-one semi-structured interviews with key stakeholders to assess DARTT intervention’s functionality, accessibility and acceptability and to refine its content. Qualitative methods were selected for their ability to generate in-depth understanding of user needs, identify contextual barriers and facilitators, and support iterative design improvements—essential when refining digital interventions in complex clinical settings. Stakeholders provided user feedback on the prototype’s components, highlighting elements they liked, found frustrating, or suggested as important additions. This early rapid testing allowed the research team to identify and address areas for improvement, ensuring the intervention was relevant and usable in clinical practice. Findings from this user engagement work are reported here.

### 4.2. Prototype Intervention

As identified in this paper’s introduction, the Phase 1 prototype intervention comprised four components (see Figure 2). These were (1) Antibiotic Tracker—an electronic prompt and review tool embedded into prescribing systems; (2) Webinar introducing DARTT and its rationale; (3) an e-training module for prescribers, pharmacists, and nurses; and (4) patient information materials—leaflets to engage patients and families in antibiotic reviews. Full details of DARTT are provided in the Appendix A.

Briefly, the Antibiotic Tracker combines an electronic antimicrobial dashboard with a decision-support tool to flag prescriptions for review, guided by principles of *environmental restructuring*, *enablement*, *and restriction* [25,26]. A traffic-light system prompts day-three antibiotic review to support de-escalation within routine workflows. Embedded hyperlinks offer quick access to local guidelines. Table 3 outlines the core interactive features; full details have been published separately [25].

The components were designed by a team of intervention development researchers with different levels of expertise (GW, NR, JS, DW, BW, KK), with input from an experienced clinician and a psychologist (DI, GG). Materials are aimed at healthcare professionals involved in everyday decision-making and/or prescription of antibiotics in acute hospitals as well as patients and their relatives to improve active antibiotic review. They were designed to be used flexibly, as standalone components or as part of an integrated toolkit, with recommendations for using all materials. The design of specific intervention components was guided by the APEASE (Acceptability, Practicability, Effectiveness, Affordability, Side-effects, Equity) criteria and coded using a Behaviour Change Technique Taxonomy (e.g., techniques to enhance capability, opportunity and motivation) [26]. BCTs were mapped to each component; for example, the Antibiotic Tracker included prompts/cues and adding objects to the environment, the e-training used instruction, habit formation and behavioural practice/rehearsal. A behavioural analysis identified the elements crucial for addressing the key behavioural and contextual issues uncovered during the planning phase, which were subsequently translated into intervention content. The outcome of this process has been reported elsewhere [25]. Integral to this work was the optimisation of DARTT, which is the primary focus of this paper.

### 4.3. Recruitment and Procedure

Ethical and management approvals for the study were obtained from the Edinburgh Napier University Research Ethics Committee (ref: SHSC/0003) and the NHS Lothian R&D Committee (ref: 2018/0007). The multi-faceted recruitment process, outlined in Figure 3, began with discussions involving clinical managers and charge nurses from six major hospitals across Scotland and England, covering critical care, A&E, and medical and surgical wards. Professional recruitment was conducted through specialist clinical networks, NHS intranet advertisements, and internal emails. For lay participants, a ‘gatekeeper’ approach was used, with an NHS Patient Advisory Service representative identifying and contacting potential participants. Additionally, Phase 1 participants who had expressed interest in future research were invited to join Phase 2 interviews. Some participants also shared study information with colleagues engaged in antimicrobial stewardship. Out of 19 volunteers who confirmed participation, 18 took part in the study. Recruitment channels included NHS Patient Advisory Services (4 participants), professional NHS networks (11 participants), and social media platforms such as Twitter/X (3 participants) (see Appendix A for details). No financial incentives, other than reimbursement for travel expenses, were provided to participants.

The researchers recruited a purposive sample of 15 healthcare professionals (HCPs) and 3 lay participants (i.e., health service users) to test the DARTT prototype (see Table 4). HCPs were recruited from four Scottish NHS Health Boards (*n* = 8) and two English Healthcare Trusts (*n* = 7) with clinical experience ranging from 2.5 to 32 years (mean 11 years, SD 8.5). Participants included junior and senior medical staff, pharmacists, and senior infection control nurses, many with direct experience in antimicrobial stewardship or working within dedicated Antimicrobial Stewardship Teams. Their roles involved optimising prescribing practices, providing clinical feedback, implementing local guidelines, and monitoring antibiotic use—offering practical insights into the integration of digital tools in stewardship efforts. HCPs were also from different disciplines, which enabled us to gain insight from diverse clinical perspectives and healthcare provider contexts, capturing a wide range of views on the prototype’s functionalities and potential usability issues from prescriber and organisational perspectives. Lay participants, recruited from Scotland (*n* = 2) and England (*n* = 1), provided feedback on the system’s accessibility and ease of use from a patient perspective. They were selected based on recent inpatient hospital experience and a willingness to engage with patient-facing information tools during the pandemic. No exclusion criteria were applied beyond age (>18 years) and the ability to provide informed consent. As data saturation was reached after interviews with three lay participants, this number was considered sufficient for evaluating the patient-facing component within the context of this study. This combined approach helped ensure the DARTT intervention met the needs of HCPs and patients.

Sampling continued until data saturation was reached and no new themes emerged. Online semi-structured interviews were conducted between 1 December 2021 and 31 January 2022, lasted 45–60 minutes, and were audio-recorded and transcribed verbatim by a professional digital audio transcription typing agency. Field notes were taken during and after interviews and included observations of participant engagement, emotional responses, and contextual details, such as technology used and environment. The topic guide was informed by the evidence-based RE-AIM Framework [9], a review of relevant literature [42,43], and the authors’ previous experience in the field. It was also pilot tested to ensure content validity (see Appendix A). Pilot testing involved three interviews, which led to refinement of several questions for clarity and inclusion of probes related to digital literacy. To build trust and ensure transparency, participants were informed of the interviewer’s background, her role in the research team, and her involvement in the intervention development as part of her doctoral work. Given the interviewer’s role as both researcher and DARTT developer, the research team implemented several safeguards, including the use of neutral questioning and ensuring participants were informed of their right to pause or withdraw at any time. All participant data, including interview recordings, transcripts, and notes, were stored securely in encrypted institutional servers. Identifiable data were anonymised during transcription, and access was restricted to authorised members of the research team.

Data collection was an iterative process, with initial interviews leading to DARTT modifications, followed by further interviews to assess these modifications. Using the PowerPoint prototyping tool, researchers sketched ideas and built a simple low-fidelity prototype (preliminary version, see Figure 4) based on the priorities identified during the intervention development (Phase 1). Low-fidelity prototypes contain basic elements of the final design, which can help visualise high-level design concepts into a tangible product and testable artefact in a quick and easy way [44]. It was a mock-up of sketches and screens of the DARTT components that allowed us to test the visual design (e.g., shapes of elements, basic visual hierarchy), content and user interaction with the Antibiotic Tracker. The key was to maintain simplicity and focus on DARTT structure and functionality, prioritising these aspects over aesthetics and style.

### 4.4. Data Analysis

Data analysis involved several concurrent components: (a) analysis of user-reported interview data using Framework Analysis and guided by the RE-AIM framework (Objective 1), and (b) user-informed modifications to DARTT, subsequently mapped onto the Behaviour Change Wheel (BCW) using the ‘*Table of Changes’* approach (Objective 2). These components were interdependent, with interview findings informing both the RE-AIM mapping and the intervention modifications. Discrepancies in coding were resolved through team discussion and comparison of excerpts until consensus was reached. Inter-coder reliability was assessed by joint review of early transcripts and regular research meetings. See Appendix A for full details of the analysis.

#### 4.4.1. Interview Data

In-depth analysis began after the first interview and continued for four weeks following the final session, spanning a 3-month period. Interviews were analysed using NVivo (v12) and Framework Analysis, which included familiarisation, developing a coding framework, indexing, charting, mapping and interpretation [45]. To support the real-world translation of the DARTT intervention, the analysis was guided by a priori issues derived from the RE-AIM framework, which is widely used and well-suited for guiding intervention adaptations [9,46]. Participants’ views and experiences of the prototype were categorised and mapped onto RE-AIM domains (see Section 2; Table 1 and Table 2). For example, *Reach* encompassed user eligibility and access, while *Adoption* included perceived barriers and benefits. These domains were refined inductively as new patterns emerged. Each RE-AIM domain was operationalised through descriptive and interpretive codes aligned with interview topics and adapted iteratively as new insights developed—for instance, codes were added to capture concerns about credibility and workload that were not originally covered. Initial coding was conducted independently by researchers GW and KK, and a final coding framework was developed collaboratively after reviewing the first four transcripts, with input from the wider research team.

#### 4.4.2. Mapping the User-Informed Modifications to DARTT

The ‘Table of Changes’ (ToC) method, adapted from Bradbury et al. [47], was used to identify user-informed modifications to improve DARTT’s acceptability and feasibility in clinical practice (see Table 2). Interview feedback informed iterative cycles of data collection, analysis, and intervention refinement. For each issue raised, the team assessed whether a feasible modification could address it. If appropriate, the change was implemented and recorded in the ToC. This structured, person-centred approach sits between participatory, theory-driven, and evidence-based models, prioritising user feedback while retaining behaviour change elements supported by theory and evidence [4,48]. Suggested changes were documented in a table detailing the proposed modification and the issues targeted, enhancing transparency and traceability. Modifications were then mapped retrospectively onto the Behaviour Change Wheel [26] to identify relevant behavioural determinants (e.g., acceptability, feasibility, motivation and engagement). Simple, uncontroversial changes—such as adding links to prescribing guidelines or reducing alert frequency—were implemented immediately.

## 5. Conclusions

This paper describes a rapid and pragmatic approach to optimising the Digital Antibiotic Review Tracking Toolkit. This was part of an iterative process of intervention planning, design, development, and refinement, where researchers integrated evidence, theoretically based behavioural analysis, and inductive qualitative research with target users, as recommended by the new Medical Research Council guidance. The optimisation phase provided detailed user feedback on the intervention and helped us identify any perceived weaknesses in the content and design. Guided by the current literature, user-centred design was used to customise the intervention to individual needs, assess technology acceptance (e.g., usability and practicality), ensure relevance to target populations, monitor progress, and incorporate strategies for long-term engagement.

Several factors were identified as key to enhancing the success of implementation efforts. These included the design of intuitive, user-friendly systems (linked to usability and acceptability), alignment with existing digital infrastructure (technological compatibility), provision of ongoing training and support (practicality), strong organisational buy-in and leadership engagement, and the presence of continuous feedback mechanisms (supporting user engagement and iterative improvement). Intervention acceptability emerged as a central theme, shaping perceptions of implementation success and influencing views on the intervention’s long-term sustainability.

The findings provide clear evidence that a structured, user-centred optimisation process can lead to meaningful refinements in digital interventions, potentially reducing antibiotic misuse through improved clinical decision-making and workflow efficiency. By explicitly detailing how user feedback can translate into practical system improvements, the study provides a concrete model for future digital health implementation, emphasising the direct impact on patient care quality and antimicrobial stewardship efforts.

Ultimately, the structured approach presented offers actionable, detailed guidance for enhancing digital intervention effectiveness, highlighting the importance of early-stage user involvement and iterative optimisation to minimise resource waste and maximise implementation success.

## Figures and Tables

**Figure 1 antibiotics-14-00660-f001:**
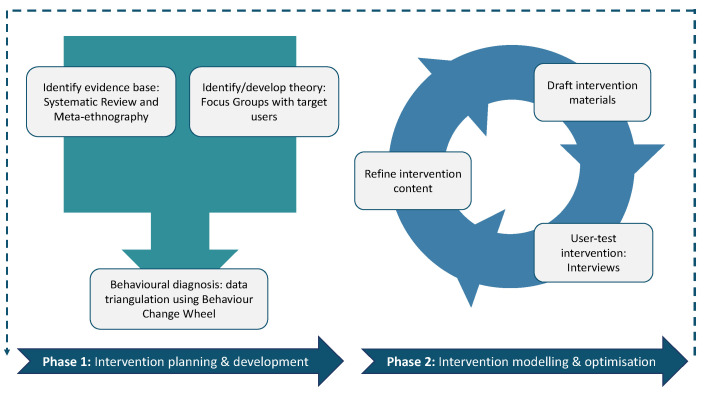
Methods applied in the development and optimisation of the DARTT intervention, with the details of intervention planning and development (Phase 1) fully reported elsewhere [25].

**Figure 2 antibiotics-14-00660-f002:**
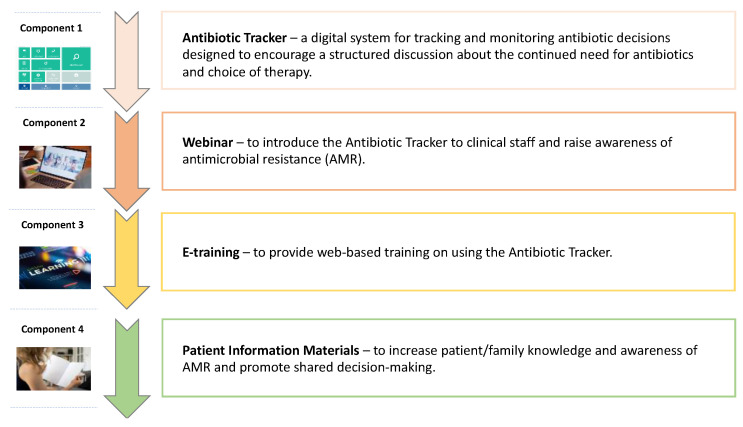
DARTT components developed in Phase 1.

**Figure 3 antibiotics-14-00660-f003:**
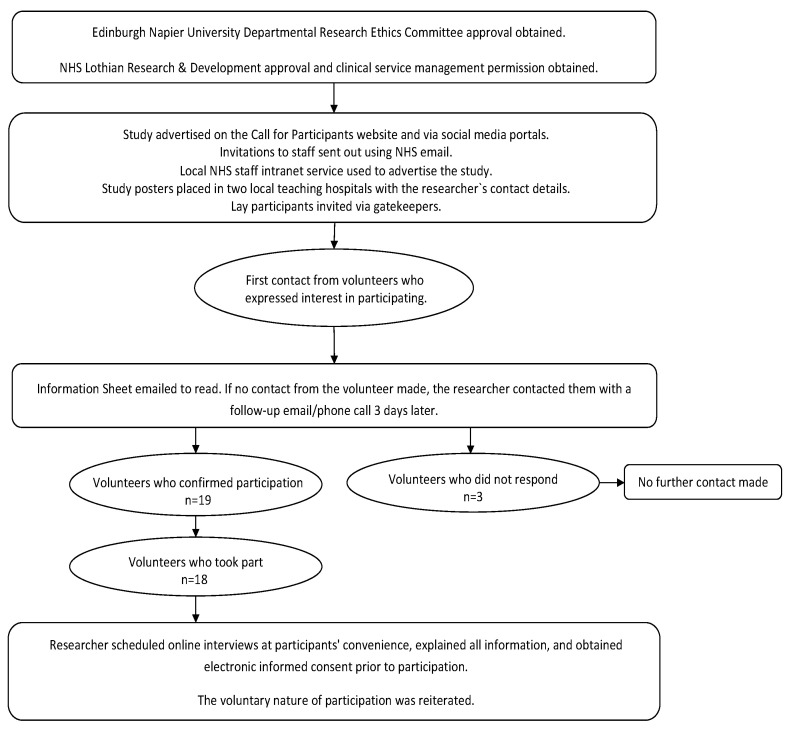
Study recruitment process.

**Figure 4 antibiotics-14-00660-f004:**
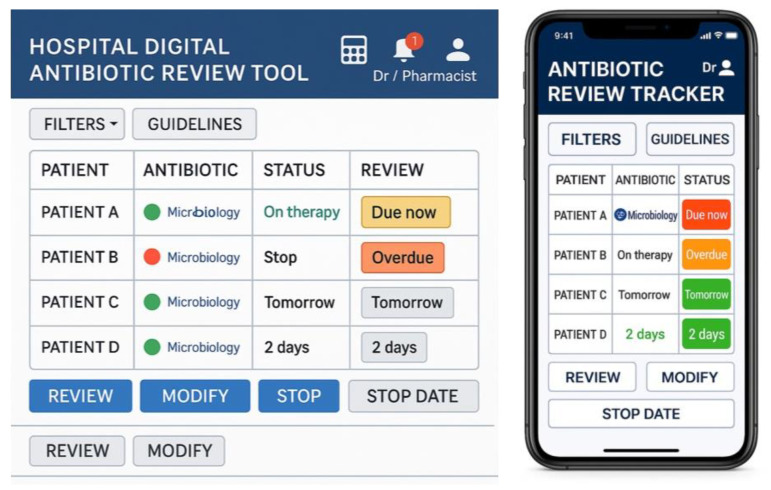
Low-fidelity prototype interfaces of the DARTT dashboard and mobile phone app used during interviews.

**Table 1 antibiotics-14-00660-t001:** User-reported themes and the corresponding RE-AIM domains.

Themes	RE-AIM Domains				
	Reach	Effectiveness	Adoption	Implementation	Maintenance
Tailoring System Functionalities and Design		✓		✓	
Bridging the Technology Gap	✓		✓		
Maintaining Organisational Leadership			✓	✓	
Lessons Learned and Sharing of Experiences		✓			✓

**Table 2 antibiotics-14-00660-t002:** An overview of key user-informed modifications to DARTT Components 1–3 and mapped to the Behaviour Change Wheel [26].

Component 1: Antibiotic Tracker				
Purpose of Change	Issues Targeted	Modifications Made	Incorporated BCTs *	Mechanisms of Action (↑ COM-B) *
Improve design and functionality.	Tracker perceived as onerous; too many alerts; risk of bypassing reviews; limited integration with other systems.	Streamlined review process; fewer required fields. Introduced authorisation codes for restricted antibiotics. Introduced prescriber dashboards with action menus. Prioritised alerts by clinical importance. Embedded microbiology result summaries. Made key fields (e.g., indication, stop date) mandatory. Enabled tracking of user activity and prior prescriptions. Integrated medical calculator and direct links to guidelines. Added option to disable reminders for certain treatments.	Environmental restructuring (dashboard, streamlined process); Prompts/cues (prioritised alerts); Adding objects to the environment (medical calculator). Feedback on behaviour (user tracking). Instruction on how to perform behaviour (links to guidelines). Behavioural regulation (mandatory fields, authorisation codes).	Physical Opportunity (easier to interact with the system); Psychological Capability (support decision-making with summaries/tools); Reflective Motivation (reduced burden increases intention to comply).
Support and feedback for healthcare professionals.	Lack of technical support and feedback mechanisms.	Created online manual and in-system help tools. Enabled direct access to IT support. Introduced feedback emails highlighting good practice. Designated ‘Antimicrobial Resistance Champion’ roles to offer peer support and individualised feedback.	Instruction on how to perform the behaviour (manuals, help tools); Social support (practical) (IT support, peer roles); Feedback on behaviour (emails); Social support (emotional) (champion roles).	Psychological Capability (through guidance and help tools); Social Opportunity (via peer and IT support); Reflective Motivation (through feedback and recognition).
**Component 2: Webinar**				
**Purpose of change**	**Issues targeted**	**Modifications made**	**Incorporated BCTs**	**Mechanisms of action (**↑ **COM-B)**
Improve engagement and integration.	Webinar too long; unclear rationale for DARRT.	Reduced webinar duration to 20 min. Added summary of DARTT’s purpose. Developed both pre-recorded and live interactive formats.	Restructuring the social environment (changing delivery format); Information about health consequences (clarifying rationale); Instruction on how to perform behaviour (engaging formats).	Physical and Social Opportunity (making engagement easier and more accessible); Reflective Motivation (clarifying purpose to enhance motivation.
**Component 3: E-training**				
**Purpose of change**	**Issues targeted**	**Modifications made**	**Incorporated BCTs**	**Mechanisms of action (**↑ **COM-B)**
Improve accessibility and practicality.	Difficulty accessing training; lack of practical elements.	Enabled remote/home access with a direct link. Embedded training in induction; made completion mandatory. Added interactive elements for hands-on learning.	Instruction on how to perform the behaviour (training content); Adding objects to the environment (access links); Habit formation (embedded in induction); Behavioural practice/rehearsal (interactive elements).	Physical Opportunity (easier access); Psychological Capability (hands-on learning); Automatic Motivation (routine through induction).

* Additional exemplar quotes supporting each modification are provided in the Appendix A (see Indexing section). The BCTs and mechanisms of action are reported separately [25].

**Table 3 antibiotics-14-00660-t003:** Core functions and user interactions with the Antibiotic Tracker.

Functionality	Description
Login and Role Recognition	Secure login with role-based access for doctors and pharmacists
Dashboard Overview	Displays patient list, antibiotic data, review due dates, and traffic light status.
Traffic Light System	Colour-coded indicators (Red = overdue, Amber = due within 24 h, Green = reviewed and up to date) for triage.
Dosing Calculator	Supports accurate, guideline-based antibiotic prescribing and reduces clinician workload.
Reminders & Alerts	Real-time prompts for due/overdue reviews, updated lab results, and non-guideline prescriptions.
Microbiology Integration	Links lab results to prescriptions; updates automatically with new sensitivities.
Guideline Access	Contextual links to local/national antibiotic prescribing guidelines.
Review Actions	Prescribers can view details, document decisions (continue, stop, change), and mark as reviewed.
Communication Tools	Shared notes and alerts notify pharmacy or microbiology teams of updates or issues.
Audit & Feedback	Personal dashboards with review compliance, decision history, and performance metrics.

**Table 4 antibiotics-14-00660-t004:** Participant characteristics.

Characteristics	Total Number of Participants (*n* = 18)
Healthcare professionals	15
Health service users	3
Gender	
Male	9
Female	9
Age range	
21–30	3
31–40	4
41–50	5
51–60	5
>60	1
Ethnicity	
White British	17
Black African	1
Current clinical position and pseudonyms (HCPs *only*)	
Consultant physician—* Robert [P1], Tom [P2]	2
Microbiology & infectious diseases consultant—Daniel [P3], Peter [P4]	2
Infectious diseases consultant—Miles [P5]	1
Medical trainees (FY1/2/Registrar)—Maya [P6], Wesley [P7], Matthew [P8]	3
Clinical pharmacist—Alex [P9], Kirstin [P10], Jane [P11]	3
Advanced nurse practitioner—Anna [P12], Caroline [P13], Hannah [P14]	3
Nurse (infection surveillance)—Mary [P15]	1
Years of clinical experience	
<5	3
5–10	2
21–30	8
>30	2
Health service users’ occupation and pseudonyms	
National Health policy officer—Sophie [P16]	1
Secondary school teacher—Veronica [P17]	1
Retired engineer—James [P18]	1

* All participants were assigned pseudonyms to protect their identities when reporting findings.

## Data Availability

The data presented in this study are available on request from the corresponding author.

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
