# Peer review of "Evaluating the Feasibility and Acceptability of a Prototype Hospital Digital Antibiotic Review Tracking Toolkit: A Qualitative Study Using the RE-AIM Framework"

_antibiotics, 2025, doi:10.3390/antibiotics14070660_

Round 1
Reviewer 1 Report
Comments and Suggestions for Authors
- The introduction provides a strong rationale for the study. However, the authors could briefly discuss the global burden of antimicrobial resistance to emphasize the importance of the intervention.
- Authors should clarify how they ensured the reliability of the coding process in NVivo.
- "Table of Changes" could be streamlined to focus on the most impactful modifications.
- Discuss challenges of scaling iterative optimization processes in larger projects in Limitations.
- Discuss how DARTT could be integrated into existing hospital workflows and electronic health record systems?
- The manuscript mentions the need for DARTT to integrate with existing hospital systems but does not specify how this integration would be achieved. For example:
- How would DARTT interact with electronic health records (EHRs)?
- Would additional training or resources be required for integration?
- The manuscript does not address the financial implications of implementing DARTT, including costs related to development, training, and ongoing maintenance.
- How will patient data be protected when using DARTT?
- The discussion is heavily skewed toward positive findings. This may create an impression of bias.
Reviewer 2 Report
Comments and Suggestions for Authors
This manuscript presents a qualitative study that optimizes a digital behavioral intervention, DARTT, designed to improve antibiotic review practices in acute hospital settings. Through stakeholder interviews, it identifies usability barriers and adapts the intervention accordingly using iterative, user-centered design principles. It is interesting for its integration of behavioral science frameworks, and its practical application of rapid qualitative methods to refine a health IT tool before trial implementation. It demonstrate that early and active stakeholder engagement can meaningfully shape the development of digital health interventions.
The following aspects should be considered:
Typos and formal errors.
- Line 58 “feasibile” → should be “feasible”
- Line 217 :“Error! Reference source not found.” → This is a placeholder referencing error (likely a missing table or figure link from MS Word or reference management software).
- Line 253 “Error! Reference source not found.” → Same issue as above. Should be corrected with the correct figure/table number.
- The reference list should formatted according to the ACS style guidelines. (See the journal's Instructions for Authors )
In the introduction
- Given that the intervention targets antibiotic use, explicitly states the severity of the AMR crisis and its implications for healthcare
- Explain with more detail why qualitative methods are particularly well-suited for optimizing digital health interventions (eg . it can provide in-depth understanding of user needs, identify contextual factors, and facilitate iterative design )
- Add a very brief sentence stating that the intervention development process was theoretically informed (e.g., by behavior change theory).
material and methods.
- Give more details on the criteria used for selecting participants. Eg: what were the varying experience levels of the HCPs? What specific types of health service users were included?
- The authors don’t provide sufficient information on the inclusion or exclusion criteria for participant selection, particularly regarding how lay participants were identified and whether their number was adequate to achieve thematic saturation.
- Indicate how many individuals were contacted through each method (e.g., how many through Twitter(now X) , how many through the NHS Patient Advisory Service)
- The manuscript provided information about their backgrounds. Pleas present if available the experience with antimicrobial stewardship
- Provide more about the pilot testing process (e.g., how many interviews, what changes were made)
- Give the range of the duration of interviews is provided (e.g., interviews ranged from 45 to 60 minutes).
- Explain what kind of information was included in the field notes (e.g., observations about the participant's demeanor, contextual information about the interview setting).
- Give more details on how the initial coding framework was developed and refined. Eg what specific deductive codes were based on the RE-AIM framework? How were disagreements in coding resolved?
- The description of the prototype intervention relies heavily on referenced figures (Figure 2),, with a limited narrative explanation of its components, which may hinder understanding for readers without access to the visuals. For instance, what exactly are the four components? One has to infer them from the narrative and Table 1.
- There is no link or example (such as figure or a screenshot) of what the mock-up or low-fidelity prototype looks like.
- Explain the procedures for keeping participants confidentiality, addressing power dynamics, and managing potential discomfort ( given the dual role of the interviewer as both researcher and developer of the intervention).
- The data analysis section notes a modification of the RE-AIM framework but does not specify which categories were altered or how the coding scheme evolved through the analysis.
- Additionally, the authors do not report on inter-coder reliability or any consensus measures taken to ensure analytic rigour.
RESULTS
- The separation between Stage1 and 2 is inconsistently maintained, creating confusion about when and how participant feedback led to concrete changes in the intervention.
- Create a clear table or matrix that directly connects participant feedback to specific modifications made to the DARTT intervention during Stage 1 and Stage 2.
- When presenting quotations, providing additional contextual information about the participant (e.g., their profession, experience level) might help the reader better understand the perspectives being shared.
DISCUSSION
- The Discussion is overly long and sometimes repetitive, particularly when summarising findings detailed extensively in the Results section. Condense the summary of results in the Discussion and focus more on interpretation, avoiding unnecessary repetition of material already detailed in the Results section.
- The Discussion includes relevant literature but tends to describe studies rather than critically engage with them. Authors should point how how the study advances, contrasts with, or fills gaps in existing knowledge about digital health intervention optimisation.
- Several important limitations are either downplayed or incompletely discussed, e.g., the lack of real-world testing of the prototype and the risk of bias due to the interviewer's dual role as a developer. These weaknesses require stronger and more transparent acknowledgment in the Discussion.
- The Discussion conflates participant-reported barriers with the authors' own hypotheses about potential challenges without clearly distinguishing empirical evidence from speculation. The authors should clearly separate participant-derived results from more speculative or recommendation-based statements (e.g., around leadership engagement and long-term sustainability).
- future research directions are vague and generic; they should be made more precise, outlining specific next steps for the DARTT intervention, such as feasibility testing protocols, outcome measures for preliminary evaluation, or plans for broader stakeholder engagement.
The paper mixed American and UK English. Eg:
“optimisation” (UK) and “behavior” (US)
“centre” (UK) and “personalized” (US) elsewhere
It will better if same variant english or American is used consistently in the manuscript.
Reviewer 3 Report
Comments and Suggestions for Authors
***The objective includes (i) identifying factors influencing the acceptability and usability of the DARTT prototype and (ii) assessing the end user views, impact, and potential refinements of DARTT's design, content, and delivery ahead of a future trial. This insight was needed to ensure that engagement with key stakeholders guided DARTT's development and optimization, aligning the intervention content with end-users needs and preferences and effectively addressing real-life challenges in conducting timely antibiotic reviews.
***According to the result, thematic analysis regarding each factor affecting the acceptability and usability of the DARTT prototype is required.
***Please review the factors regarding the interview questions. Please use interview questions based on the previous studies in this study and test the validity of the content of the interview questions.
***According to the conceptual framework, one figure is required for the factors affecting acceptability and usability (the first research objective). Another figure is required for the second research objective.
***This topic is entitled "Using Stakeholder Feedback to Optimize a Digital Behavioral Change Intervention to Promote Active Antibiotic Review in Hospitals: A Qualitative Study." The researchers use the term "hospitals," indicating that more than one hospital is included in this study. Please justify the study's setting (population) and sampling technique. The sample totals 18, including 17 British and 1 Black African; it seems like a purposive sampling. Please justify its selection criteria.
***The sampling technique includes two steps; the first step is to select the hospitals to represent the study's setting. The second step is to choose the interviewee; please give more details.
***According to the results, please add more details on the date and time of the interviews for the two research objectives and present a thematic analysis regarding the factors and the outcome with some interview answers.
***The RE-AIM (Reach, Effectiveness, Adoption, Implementation, and Maintenance) informed the data analysis; please add the details of this analysis.
***Figure 2: DARTT components comprise (1) antibiotic tracker, (2) webinar, (3) E-training, and (4) patient information material. Please justify in Figure 2 that DARTT (model) components are the results of the first or second research objective.
***Please justify the results of the first or second research objectives regarding APEASE (Acceptability, Practicability, Effectiveness, Affordability, Side Effects, Equity).
***Please avoid using we, but use researchers instead.
***Table 1 shows three components, but Figure 2 shows four components of the DARTT model; please check.
***Finally, please discuss and conclude by following the two research objectives.
***Please check the thematic analysis (themes 1-4) to see if they are following the two research objectives regarding the DARTT model or the APEASE framework.
***The conclusion is necessary to explain how a strategic planner could improve the outcome based on the theories or variables (factors) in this study; please check the abstract and the conclusion. Please justify the factors and the outcome variable in this study.
Reviewer 4 Report
Comments and Suggestions for Authors
I have read the manuscript and found the topic to be interesting; however, there are several issues that need to be addressed by the authors.
Major Issues:
The current title partially reflects the scope of the study but falls short of fully capturing its objectives and methodological framework. While it mentions key elements, it omits crucial aspects that define the study’s core purpose such as the feasibility, acceptability, and usability assessment of the Digital Antibiotic Review Tracking Toolkit (DARTT) guided by the RE-AIM framework. The term “optimise” may also mislead readers into assuming a post-trial refinement process, whereas this is a pre-trial formative evaluation. Additionally, the absence of the intervention’s name (DARTT) and the RE-AIM framework reduces the title’s clarity and relevance for readers interested in implementation science. A more accurate and informative title is recommended to better reflect the full scope and contribution of the study. Suggested title is: “Evaluating the Feasibility and Acceptability of a Prototype Digital Antibiotic Review Tool in Hospitals: A Qualitative Study Using the RE-AIM Framework”
Abstract:
- The phrase “to understand the acceptability and usability…” does not fully convey that the aim was to evaluate the intervention's feasibility across multiple implementation dimensions (Reach, Effectiveness, Adoption, Implementation, and Maintenance).
- The term “optimisation” is used, but the abstract does not explain what “optimisation” entailed or that it was guided by a structured implementation science framework.
- Although RE-AIM is mentioned, it is not briefly explained or framed as a guiding structure for data analysis. This may confuse readers unfamiliar with the term.
- It should specify that the study was conducted among healthcare professionals and service users in hospital settings in the UK.
- The conclusion is vague about the significance of the findings. It mentions that the study is "novel" and "efficient" but does not clearly state how the findings could impact future research or digital health implementation.
Introduction:
- The link between antibiotic misuse, hospital workflow inefficiencies, and the role of DARTT could be made more explicit to clarify the public health significance.
- Sentences like "This novel study..." or "This paper presents an example of..." feel more appropriate for the Discussion or Abstract.
- While DARTT is mentioned, its features and intended use are not described until later in the manuscript. This weakens the reader’s early understanding of what the intervention is and why it matters.
- The core problem (i.e., low uptake and inconsistent implementation of antibiotic review processes) is not clearly defined early on.
- The introduction blends the background, research gap, and rationale without clear transitions.
Method and tool:
- Several points in the methods section refer to Figure 1, Table 1, and Table of Changes, but these are either duplicated or flagged as “Error! Reference source not found”—this undermines credibility and flow.
- The DARTT components (Antibiotic Tracker, Webinar, e-training, Patient Info) are mentioned, but their structure, functions, and delivery mechanisms are not consistently or clearly presented in this section - Elaborate briefly on what each DARTT component entails, especially the Antibiotic Tracker and how users interacted with it.
- The relationship between the Behaviour Change Techniques (BCTs) used and each DARTT component is vague – need more explanation
- While NVivo and Framework Analysis are mentioned, how codes were generated and how RE-AIM dimensions were operationalised during coding should be described in more detail.
- The use of the Table of Changes (ToC) is mentioned as part of the rapid optimisation process, but the step-by-step process and validation of changes are unclear - Ensure clarity in how feedback led to iterative design changes perhaps a visual or table linking participant feedback to tool modifications.
- While COREQ adherence and triangulation are mentioned, the section could benefit from briefly listing measures of credibility, dependability, confirmability, and transferability here (instead of solely in the discussion).
Results:
- Mentions of Table 1 (participant characteristics) and the Table of Changes are referenced, but no such tables are visible. Instead, placeholder errors like “Error! Reference source not found” appear.
- While the themes do reflect RE-AIM dimensions, they are not clearly labeled as such (e.g., Theme 1: Tailoring functionality – aligns with ‘Implementation’ and ‘Effectiveness’). This weakens the connection between the framework and findings for readers unfamiliar with RE-AIM.
- Some participant quotes are too lengthy or redundant, which affects readability and focus. A few insights are repeated across multiple themes (e.g., feedback mechanisms and workflow integration) - condense overly long participant quotes and avoid redundancy.
Discussion, limitations and conclusion:
- Despite using RE-AIM in the methodology and analysis, the discussion does not systematically interpret findings through each of its five dimensions (Reach, Effectiveness, Adoption, Implementation, Maintenance).
- Several ideas (e.g., digital literacy concerns, need for leadership support) are repeated multiple times.
- While there is brief mention of implications for policy and future research, there is no clear roadmap for the next steps (e.g., feasibility trial, cost-effectiveness assessment, scale-up strategy).
- The limitation is well written.
- The conclusion is too general and lacks actionable next steps.
Reference list:
- Some references cited in the manuscript are either incomplete or not listed correctly. For example: "Error! Reference source not found.” appears multiple times in the manuscript, indicating missing or improperly linked in-text citations.
- Some references include DOI hyperlinks, others only plain DOIs or none at all.
Round 2
Reviewer 2 Report
Comments and Suggestions for Authors
Dear authors, thanks for incorporating my points into the manuscript. In my opinion it can be published as it is, in its present form.
Author Response
Thank you very much for taking the time to review our manuscript. We are pleased to hear that the revised version has addressed your concerns and that you found the revised manuscript publishable in its present form.
Reviewer 3 Report
Comments and Suggestions for Authors
***Methods in the abstract required for data collection and analysis. The study's setting (population), sample, sample size, and data analysis are still needed in the abstract and the methodology section.
***Please justify that the respondents (n=18) were from several organizations (how many?) or only one.
***HCPs were recruited from four Scottish NHS Health Boards and two English Healthcare Trusts. Does it mean that the study's setting represents Scotland and English countries? If yes, please discuss and recommend further studies.
***The sample was 15 healthcare professionals and 3 health service users: please discuss how the perceptions differ among the two groups.
***Line 571: The researchers recruited a purposive sample of 15 healthcare professionals (HCPs), but Table 2 shows that the respondents were 18; please verify and revise accordingly.
***Table 3 needs references based on previous studies to support this.
***Please move headings 3.1 and 3.2 after the conclusion (or add them in the conclusion).
***Methodology is still unclear and needs information, as mentioned. Additionally, please include interview questions based on previous studies to support the related theories and variables in this study.***
Reviewer 4 Report
Comments and Suggestions for Authors
Thank you for sharing the revised version of the manuscript. The authors have comprehensively addressed both major and minor concerns, making it overall well-structured, scientifically robust, and methodologically sound. I have no further concerns and recommend the manuscript for acceptance and publication. Congratulations to the authors on their thorough and thoughtful revisions.
Author Response
Thank you very much for the time to review our manuscript. We are pleased to hear that the revised version has addressed your concerns and that you found the manuscript to be scientifically robust and methodologically sound.
Round 3
Reviewer 3 Report
Comments and Suggestions for Authors
***Please avoid using "we" but use "researchers" instead.
***Please ensure that the data collection period has been provided.
***The interview questions based on the previous studies to support the theories and variables in this study are not provided.
***Line 501: The intervention design was theoretically informed by behaviour change theory. Please justify this theory that is related to this study in the discussion based on previous studies.
***Data analysis using NVivo (v12); please show some of NVivo's results.
***According to the conclusion: Strategic planners can enhance outcomes; please justify the outcome. *** by addressing these specific factors: designing intuitive, user-friendly systems (usability and acceptability), ensuring compatibility with existing digital infrastructure (technological compatibility), providing ongoing training and support (practicality), fostering organisational buy-in and committed leadership (organisational support and leadership engagement), and maintaining continuous feedback mechanisms (user engagement and iterative improvement). The primary outcome variable, intervention acceptability, significantly influences implementation success and long-term sustainability. Please ensure that all mentioned have been added in the results and discussion. This statement is the conclusion from the interview results or the recommendation for further studies; please identify which is the related tables for this conclusion.
***If Table 3 includes the variables, please add them to the conclusion.
***It is still unclear what the related theories, factors, and outcomes are. Please revise the discussion and conclusion. Any discussion and conclusion require the study's results to be supported.
